# CZ-GEM: A FRAMEWORK FOR DISENTANGLED REPRESENTATION LEARNING

## ABSTRACT

Learning disentangled representations of data is one of the central themes in unsupervised learning in general and generative modeling in particular. In this work, we tackle a slightly more intricate scenario where the observations are generated from a conditional distribution of some known control variate and some latent noise variate. To this end, we present a hierarchical model and a training method (CZ-GEM[1]) that leverages some of the recent developments in likelihood-based and likelihood-free generative models. We show that CZ-GEM introduces the right inductive biases that ensure the disentanglement of the control from the noise variables, while also keeping the components of the control variable disentangled. This is achieved without compromising on the quality of the generated samples. Our approach is simple, general, and can be applied both in the supervised and unsupervised settings.

## 1 INTRODUCTION

Consider the following scenario: a hunter-gatherer walking in the African Savannah some 50,000 years ago notices a lioness sprinting out of the bush towards her. In a split second, billions of photons reaching her retinas carrying an enormous amount of information: the shade of the lioness' fur, the angle of its tail, the appearance of every bush in her field of view, the mountains in the background and the clouds in the sky. Yet at this point there is a very small number of attributes which are of importance: the type of the charging animal, its approximate velocity and its location. The rest are just details. The significance of the concept that the world, despite its complexity, can be described by a few explanatory factors of variation, while ignoring the small details, cannot be overestimated. In machine learning there is a large body of work aiming to extract low-dimensional, interpretable representations of complex, often visual, data. Interestingly, many of the works in this area are associated with developing generative models. The intuition is that if a model can generate a good approximation of the data then it must have learned something about its underlying representation. This representation can then be extracted either by directly inverting the generative process (Srivastava et al., 2019b) or by extracting intermediate representations of the model itself (Kingma & Welling, 2014; Higgins et al., 2017). Clearly, just learning a representation, even if it is low-dimensional, is not enough. The reason is that while there could be many ways to compress the information captured in the data, allowing good enough approximations, there is no reason to a priori assume that such a representation is interpretable and disentangled in the sense that by manipulating certain dimensions of the representation one can control attributes of choice, say the pose of a face, while keeping other attributes unchanged. The large body of work on learning disentangled representations tackles this problem in several settings; fully supervised, weakly supervised and unsupervised, depending on the available data (Tran et al., 2018; Reed et al., 2014; Jha et al., 2018; Mathieu et al., 2016; Higgins et al., 2017; Chen et al., 2018; Kim & Mnih, 2018; Chen et al., 2016; Nguyen-Phuoc et al., 2019; Narayanaswamy et al., 2017). Ideally, we would like to come up with an unsupervised generative model that can generate samples which approximate the data to a high level of accuracy while also giving rise to a disentangled and interpretable representation. In the last decade two main approaches have captured most of the attention; Generative Adversarial Networks (GANs) and Variational Auto-Encoders (VAEs). In their original versions, both GANs (Goodfellow et al., 2014) and VAEs (Kingma & Welling, 2014) were trained in an unsupervised manner and

---

[1] CZ-GEM: CZ-Generative Model

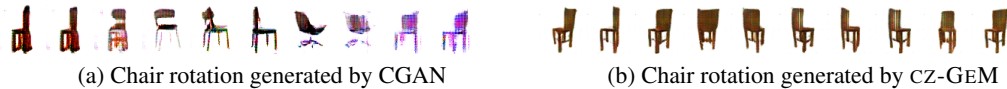

(a) Chair rotation generated by CGAN          (b) Chair rotation generated by CZ-GEM

Figure 1: Changing the azimuth of chairs in CGAN and CZ-GEM while holding $Z$ constant. Unlike CZ-GEM, $C$ and $Z$ are clearly entangled in CGAN as changing $C$ also changes the type of chair even though $Z$ is held constant.

gave rise to entangled representations. Over the years, many methods to improve the quality of the generated data as well as the disentanglement of the representations have been suggested (Brock et al., 2018; Kingma & Dhariwal, 2018; Nguyen-Phuoc et al., 2019; Jeon et al., 2018). By and large, GANs are better than VAEs in the quality of the generated data while VAEs learn better disentangled representations, in particular in the unsupervised setting.

In this paper, we present a framework for disentangling a small number of control variables from the rest of the latent space which accounts for all the additional details, while maintaining a high quality of the generated data. We do that by combining VAE and GAN approaches thus enjoying the best of both worlds. The framework is general and works in both the supervised and unsupervised settings. Let us start with the supervised case. We are provided with paired examples $(x, c)$ where $x$ is the observation and $c$ is a control variate. Crucially, there exists a one-to-many map from $c$ to the space of observations, and there are other unobserved attributes $z$ (or noise) that together completely define $x$. For instance, if $x$ were an image of a single object, $c$ controls the orientation of the object relative to the camera and $z$ could represent object identity, texture or background.

Our goal is to learn a generative model $p_\theta(x|c, z)$ that fulfills two criteria:

1. $\int_{\mathbf{z}} \mathbf{p}_\theta(\mathbf{x}|\mathbf{c}, \mathbf{z})\mathbf{p}(\mathbf{c})\mathbf{p}(\mathbf{z})\mathbf{dz}$ **matches the joint distribution** $\int_{\mathbf{z}} \mathbf{p}(\mathbf{x}|\mathbf{c}, \mathbf{z})\mathbf{p}(\mathbf{c})\mathbf{p}(\mathbf{z})\mathbf{dz}$**:** If we were learning models of images, we would like the generated images to look realistic and match the true conditional distribution $p(x|c)$.

2. **The posterior is factorized** $\mathbf{p}(\mathbf{c}, \mathbf{z}|\mathbf{x}; \theta) = \mathbf{p}(\mathbf{c}|\mathbf{x}; \theta)\mathbf{p}(\mathbf{z}|\mathbf{x}; \theta)$**:** We would like the control variate to be disentangled from the noise. For example, changing the orientation of the object should not change the identity under our model.

This problem setup can occur under many situations such as learning approximate models of simulators, 3D reconstructions, speaker recognition (from speech), and even real-world data processing in the human brain as in the hunter-gatherer example above.

We argue that a naive implementation of a graphical model as shown in Figure 2 (left), e.g. by a conditional GAN (Mirza & Osindero, 2014), does not satisfy Criterion 2. In this model, when we condition on $x$, due to d-separation, $c$ and $z$ could become dependent, unless additional constraints are posed on the model. This effect is demonstrated in Figure 1(a). To overcome this we split the generative process into two stages by replacing $C$ with a subgraph $(C \rightarrow Y)$ as shown in Figure 2 (center). First, we generate a crude approximation $y$ of the data which only takes $c$ into account. The result is a blurry average of the data points conditioned on $c$, see Figure 2 (right). We then feed this crude approximation into a GAN-based generative model which adds the rest of the details conditioned on $z$. We call this framework CZ-GEM. The conditioning on $z$ in the second stage must be done carefully to make sure that it does not get entangled with $y$. To that end we rely on architectural choices and normalization techniques from the style transfer literature (Huang & Belongie, 2017) [2]. The result is a model which generates images of high quality while disentangling $c$ and $z$ as can be clearly seen in Figure 1(b). Additionally, in the unsupervised setting, when the labels $c$ are not available, $(C \rightarrow Y)$ can be realized by $\beta$-VAE, a regularized version of VAE which has been shown to learn a disentangled representation of its latent variables (Higgins et al., 2017; Burgess et al., 2018). In Section 3 we provide implementation details for both the supervised and unsupervised versions.

We summarize our two main contributions:

---

[2] Indeed the mapping from $Y$ to $X$ conditioned on $Z$ can be viewed as a general form of style transfer.

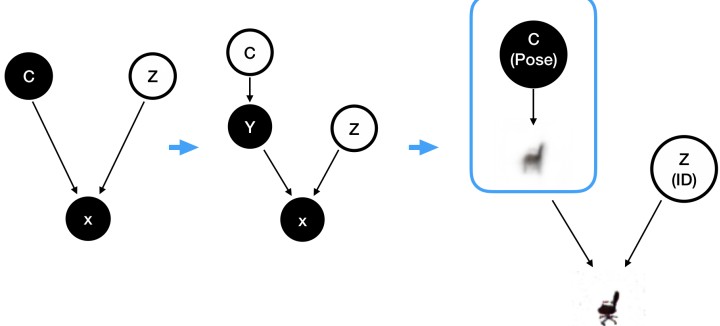

Figure 2: On the left, a conditional GAN (CGAN) model. CZ-GEM in the middle replaces node $C$ with a subgraph $(C \rightarrow Y)$ that is trained independently of the rest of the model. This subgraph learns to only partially render the observation. As such, $Z$ comes at a later stage of the rendering pipeline to add details to $Y$. As an example, consider the rightmost graph where the observation is made up of different types of chairs in different poses. Let the pose be controlled by $C$ and the type (Identity) be explained by $Z$. Then in step one of CZ-GEM we learn the pose relationship between $C$ and $X$ via the subgraph, giving rise to a blurry chair in the correct pose. Once the pose is learned, in the second step, the approximate rendering $Y$ is transformed into $X$ by allowing $Z$ to add identity related details to the blurry image.

1. **Architectural biases:** We break down the architecture to model an intermediate representation that lends itself to interpretability and disentanglement, and then (carefully) use a GAN based approach to add the rest of the details, thus enjoying a superior image generation quality compared to VAEs.

2. **Unsupervised discovery:** We show that our model can be combined easily with common methods for discovering disentangled representations such as $\beta$-VAE to extract $c$ and treat them as labels to generate images that do not compromise on generative quality.

## 2 PRELIMINARIES

### 2.1 GENERATIVE ADVERSARIAL NETWORKS

Generative adversarial networks (GAN) (Goodfellow et al., 2014) represent the current state of the art in likelihood-free generative modeling. In GANs, a generator network $G_\theta$ is trained to produce samples that can fool a discriminator network $D_\omega$ that is in turn trained to distinguish samples from the true data distribution $p(x)$ and the generated samples $G_\theta(z)|z \sim p_z(z)$. Here, $p_z$ is usually a low dimensional easy-to-sample distribution like standard Gaussian. A variety of tricks and techniques need to be employed to solve this min-max optimization problem. For our models, we employ architectural constraints proposed by DC-GAN (Radford et al., 2015) that have been widely successful in ensuring training stability and improving generated image quality.

Conditional GANs (CGAN) (Mirza & Osindero, 2014) adapt the GAN framework for generating class conditional samples by jointly modeling the observations with their class labels. In CGAN, the generator network $G_\theta$ is fed class labels $c$ to produce fake conditional samples and the discriminator $D_\omega$ is trained to discriminate between the samples from the joint distribution of true conditional and true labels $p(x|c)p(c)$ and the fake conditional and true labels $p_\theta(x|c)p(c)$.

While not the main focus of this paper, we present a novel information theoretic perspective on CGANs. Specifically, we show that CGAN is trained to maximize a lower-bound to the mutual information between the observation and its label while simultaneously minimizing an upper-bound to it. We state this formally:

**Lemma 1** (Information-theoretic interpretation of CGAN). *Given $(x, c) \sim p(x, c)$, CGAN learns the distribution $p_\theta(x) = G_\theta(x)$ by training a discriminator $D_\omega$ to approximate the log-ratio of the*

*true and generated data densities i.e. $D_\omega \approx \log p(x,c)/p_\theta(x,c)$ in turn minimizing the following*

$$\min_\theta \mathbb{E}_{p_\theta(x|c)p(c)}\big[-D_\omega\big] \approx \min_\theta \big(\mathcal{I}_{g,\theta}(x,c) + \mathbb{E}_{q(c|x,\theta)}KL(p_\theta(x)\|p(x))\big)+$$

$$\max_\theta \big(\mathcal{I}_{g,\theta}(x,c) - \mathbb{E}_{p_\theta(x)}KL[q(c|x,\theta)\|p(c|x)]\big)$$

$$=\min_\theta \mathcal{I}_{g,\theta}^{UB}(x,c) - \mathcal{I}_{g,\theta}^{LB}(x,c).$$

*where $\mathcal{I}_{g,\theta}(x,c)$ is the generative mutual information and $q(c|x,\theta)$ is the posterior under the learned model.*

The detailed derivation is provided in Appendix A.1. Notice that at the limit, the model learns exactly the marginal distribution of $x$ and the posterior $q(c|x)$ and the KL terms vanish.

## 2.2 VARIATIONAL AUTOENCODER

Variational autoencoders (VAE) represent a class of likelihood-based deep generative models that have recently been extensively studied and used in representation learning tasks (Higgins et al., 2017; Burgess et al., 2018; Chen et al., 2018). Consider a latent variable model where observation $X$ is assumed to be generated from some underlying low-dimensional latent feature space $Z$. VAE models learn the conditional distribution $p(x|z)$ using a deep neural network (parameterized by $\theta$) called decoder network. It uses another deep neural network (parameterized by $\phi$), called encoder to model the posterior distribution $p(z|x)$. The encoder and decoder networks are trained using amortized variational inference (Kingma & Welling, 2014) to maximizes a variational lower-bound to the evidence likelihood (ELBO). Recently, Higgins et al. (2017) showed that by regularizing the variational posterior approximation of $p(z|x)$ to be close to the prior distribution $p(z)$ in KL-divergence, the model is encouraged to learn disentangled representations. I.e. the model learns a posterior distribution that is factorized over the dimensions. They call their model $\beta$-VAE. We note that information bottleneck based methods for disentangled representation learning, such as $\beta$-VAE, severely compromise the generative quality.

## 2.3 NORMALIZATION

Batch-Normalization (BN) (Ioffe & Szegedy, 2015) plays a crucial role in ensuring the stability of GAN training Radford et al. (2015). However, as we discuss in Section 3, it is not suitable for our purposes. Recently, it has been shown that Instance Normalization (IN) Ulyanov et al. (2016) and its variant Adaptive Instance Normalization (AdaIN) Huang & Belongie (2017) can be particularly useful for image generation and stylization. IN normalizes each convolutional channel per training sample, while AdaIN modifies this normalization to be a function of an additional variable $z$ (usually style in style transfer). The final transformation applied by AdaIN is:

$$AdaIN(x,z) = \gamma(z)\Big(\frac{x - \mu(x)}{\sigma(x)}\Big) + \beta(z) \tag{1}$$

where $\mu(x) = \frac{1}{HW}\sum_{h,w} x_{nhwc}$ and $\sigma(x) = \sqrt{\frac{1}{HW}\sum_{h,w}(x_{nhwc} - \mu(x))^2 + \epsilon}$. $\gamma(z)$ and $\beta(z)$ are learned functions of $z$ that could be parameterized by a neural network, usually a fully connected layer.

## 3 CZ-GEM

In Section 1, we provided a high level description of our approach. We will now provide a detailed description of how the two components of CZ-GEM, subgraph $C \rightarrow Y$ and the conditional generative models $(Y,Z) \rightarrow X$ are implemented and trained in practice. Figure 3 provides an implementation schematic of our proposed framework.

### 3.1 SUB-GRAPH LEARNING

If $C$ is known a priori then learning the subgraph $C \rightarrow Y$ reduces to the regression problem that minimizes $||x_c - y_c||^2$. In practice, since our observations are images, this subgraph is realized using

a deep transposed-convolution based decoder network and is trained to learn the map between $C$ and $Y$. This is similar to the recent work of Srivastava et al. (2019b). We emphasize that this network is trained independently of the rest of the model.

Subgraph $C \rightarrow Y$ not only improves disentanglement and interpretability in CZ-GEM but also allows for unsupervised discovery of generative factors when $C$ is not available in the dataset. In such cases, subgraph $C \rightarrow Y$ is realized as a variational autoencoder. This allows us to use any state-of-the-art unsupervised disentangled representation learning method such as Higgins et al. (2017); Burgess et al. (2018); Chen et al. (2018); Kim & Mnih (2018); Chen et al. (2016) to discover disentangled generative control factors. In our implementation we use $\beta$-VAE (see Section 2.2). One drawback of these information bottleneck based methods is that they compromise on the generative quality. This is where the GAN likelihood-free approach in the second stage comes into play. In fact, even if the output of the first stage (i.e. the intermediate image $Y$ in Figure 2) is of very low generative quality, the final image is of high quality since the second stage explicitly adds details using a state-of-the-art GAN method. In Section 5 we show how a simple VAE with a very narrow information bottleneck

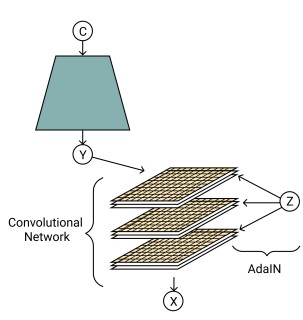

Figure 3: Schematic for CZ-GEM

(2-6 dimensions) can be used within CZ-GEM to discover $C$ in an unsupervised fashion without compromising on generation quality.

## 3.2 Adding Details

Vanilla GANs can only model the marginal data distribution i.e. they learn $p_\theta$ to match $p_x$ and in doing so they use the input to the generator ($G_\theta$) only as a source of stochasticity. Therefore we start with a conditional GAN model instead, to preserve the correspondence between $Y$ and $X$. As shown in section 2.1, this framework trains $G_\theta$ such that the observation $X$ is maximally explained by the conditioning variable $Y$. One major deviation from the original model is that the conditioning variable in our case is the same type and dimensionality as the observation. That is, it is an image, albeit a blurry one. This setup has previously been used by Isola et al. (2017) in the context of image-to-image translation.

Incorporating $Z$ requires careful implementation due to two challenges. First, trivially adding $Z$ to the input along with $Y$ invokes d-separation and as a result $Y$ and $Z$ can get entangled. Intuitively, $Z$ is adding high level details to the intermediate representation $Y$. We leverage this insight as an inductive bias, by incorporating $Z$ at higher layers of the network rather than just feeding it as an input to the bottom layer. A straightforward implementation of this idea does not work tough. The reason is that BatchNorm uses batch-level statistics to normalize the incoming activations of the previous layer to speed up learning. In practice, mini-batch statistics is used to approximate batch statistics. This adds internal stochasticity to the generator causing it to ignore any externally added noise, such as $Z$. An elegant solution to resolve this second challenge comes in the form of adaptive instance normalization (see Section 2.3). It not only removes any dependency on the batch-statistics but also allows for the incorporation of $Z$ in the normalization process itself. For this reason, it has previously been used in style transfer tasks (Huang & Belongie, 2017). We replace all instances of BatchNorm in the generator with Adaptive InstanceNorm. We then introduce $Z$ to the generative process using equation 1. $\gamma(z)$ and $\beta(z)$ are parameterized as a simple feed-forward network and are applied to each layer of AdaIN in the generator.

## 4 Related Work

Disentangled representation learning has been widely studied in recent years, both in the supervised and unsupervised settings. In supervised cases, works such as Tran et al. (2018); Reed et al. (2014); Bao et al. (2018); Jha et al. (2018); Mathieu et al. (2016); Szabó et al. (2017); Isola et al. (2017); Kulkarni et al. (2015); Narayanaswamy et al. (2017) have used the provided labels in the dataset or other form of weak supervision to promote disentanglement in the learned representation. In unsupervised learning most methods such as Higgins et al. (2017); Burgess et al. (2018); Chen et al. (2018; 2016); Esmaeili et al. (2018); Jeon et al. (2018) rely on creating an information bottleneck

to squeeze out representations that are statistically independent in its components. Such models often have to sacrifice on the generative capacity of the method in order to learn a factorized latent representation.

Recently, Locatello et al. (2018) has emphasized the use of inductive biases and weak supervision instead of fully unsupervised methods for disentangled representation learning. Nguyen-Phuoc et al. (2019) and Sitzmann et al. (2019) have successfully shown that including inductive biases, respectively an explicit 3D representation, leads to better performance. Their inductive bias comes in the form of learned 3D transformation pipeline. In comparison, CZ-GEM is much simpler and smaller in design and applies to the general setting where the data is determined by control and noise variables. In addition, it can be used in both supervised and unsupervised setting and does not rely on the knowledge of 3D transformations.

Manually disentangled generative models like the 3D morphable model (Blanz & Vetter, 1999) have been built for faces. They are powerful in terms of generalization but there is a big gap between those synthetic images and real-world face images. In addition, those models are built highly supervised from 3D scans and the approach is limited by the correspondence assumption which does not scale to more complex objects like chairs (Egger et al., 2019). We use a 3D morphable model to generate our synthetic face dataset and show that we can disentangle pose variation from synthetic and real 2D images.

# 5 EXPERIMENTS

In this section, we provide a comprehensive set of quantitative and qualitative results to demonstrate how CZ-GEM is clearly able to not only disentangle $C$ from $Z$ in both supervised and unsupervised settings but also ensure that independent components of $C$ stay disentangled after training. Additionally, we show how in unsupervised settings CZ-GEM can be used to discover disentangled latent factors when $C$ is not explicitly provided.

We evaluate CZ-GEM on a variety of image generation tasks which naturally involve observed attributes $C$ and unobserved attributes $Z$. To that end, we generate three 3D image datasets of faces, chairs, and cars with explicit control variables. Chairs and cars datasets are derived from ShapeNet (Chang et al., 2015). We sample 100k images from the full yaw variation and a pitch variation of 90 degrees. We used the straight chair subcategory with 1968 different chairs and the sedan subcategory with 559 different cars. We used Blender to render the ShapeNet meshes scripted with the Stanford ShapeNet renderer. For faces, we generated 100k images from the Basel Face Model 2017 (Gerig et al., 2018). We sample shape and color (first 50 coefficients), expressions (first 5 coefficients), pose (yaw -90 to 90 degrees uniformly, pitch and roll according to a Gaussian with variance of 5 degrees) and the illumination from the Basel Illumination Prior (Egger et al., 2018). For the generation of the faces dataset, we use the software provided by Kortylewski et al. (2019). For the stated datasets we have complete access to $C$, but we also include unsupervised results on celebA (Liu et al., 2015) with unconstrained real images. All our datasets are built from publicly available data and tools.

We use the DCGAN architecture (Radford et al., 2015) for all neural networks involved in all the experiments in this work and provide a reference implementation with exact architecture and hyperparameter settings at *https://github.com/AnonymousAuthors000/CZ-GEM*.

## 5.1 SUPERVISED SETTING

In the supervised setting we compare CZ-GEM to CGAN. We quantitatively compare the two methods to ensure that independent components of $C$ stay disentangled post learning. Furthermore, we qualitatively compare their abilities to disentangle $C$ and $Z$. And finally, we compare the quality of the samples that the models generate. For chairs and cars, $C$ contains only the pose variables and all other variations are explained by $Z$. For faces, $C$ contains in addition to pose the first 4 principal directions of shape variations.

We start by quantitatively evaluating how well the information about the control variable is maintained in the generated image. To that end we train a deep regression model $f(x)$ to predict the control variate $C$ from the dataset $X$. We then compute the mean-squared error (MSE) of this regressor on the generated data $||c - f(G_\theta(c, z))||_2^2$. In Table 1 we report these numbers for both models on all

Table 1: Disentangled representation learning performance. Lower is better. We report MSE for all the models and the data for reference.

| Dataset | Real data | CGAN | CZ-GEM |
|---|---|---|---|
| Face | 0.010 | **0.010** | 0.010 |
| Chair | 0.027 | 0.067 | **0.060** |
| Car | 0.016 | **0.032** | 0.037 |

Table 2: Inception score. Higher is better. Inception score of real data is reported for reference.

| Dataset | Real data | CGAN | CZ-GEM |
|---|---|---|---|
| Face | 2.49 | 1.90 | **2.28** |
| Chair | 3.55 | 2.49 | **3.50** |
| Car | 3.18 | 2.78 | **3.41** |

three datasets. We also include the training error (i.e. the MSE of the regressor on the real data) for comparison. The results show that CGAN and CZ-GEM are comparable in preserving the label information in the generated data, but as we show below, only CZ-GEM does that while ensuring that $C$ and $Z$ remain disentangled.

To qualitatively evaluate the level of disentanglement between $C$ and $Z$, we vary each individual dimension of $C$ over its range while holding $Z$ constant. We plot the generated images for both models on car and chair datasets in Figure 4. Notice that CZ-GEM allows us to vary the control variates without changing the identity of the object, whereas CGAN does not. In addition, we find that for CGAN, the noise $Z$ provides little to no control over the identity of the chairs. This is potentially due to the internal stochasticity introduced by the BatchNorm. The last rows for the CZ-GEM figures provide the visualization of $Y$. It can be seen how $Y$ is clearly preserving $C$ (pose information) but averaging the identity related details.

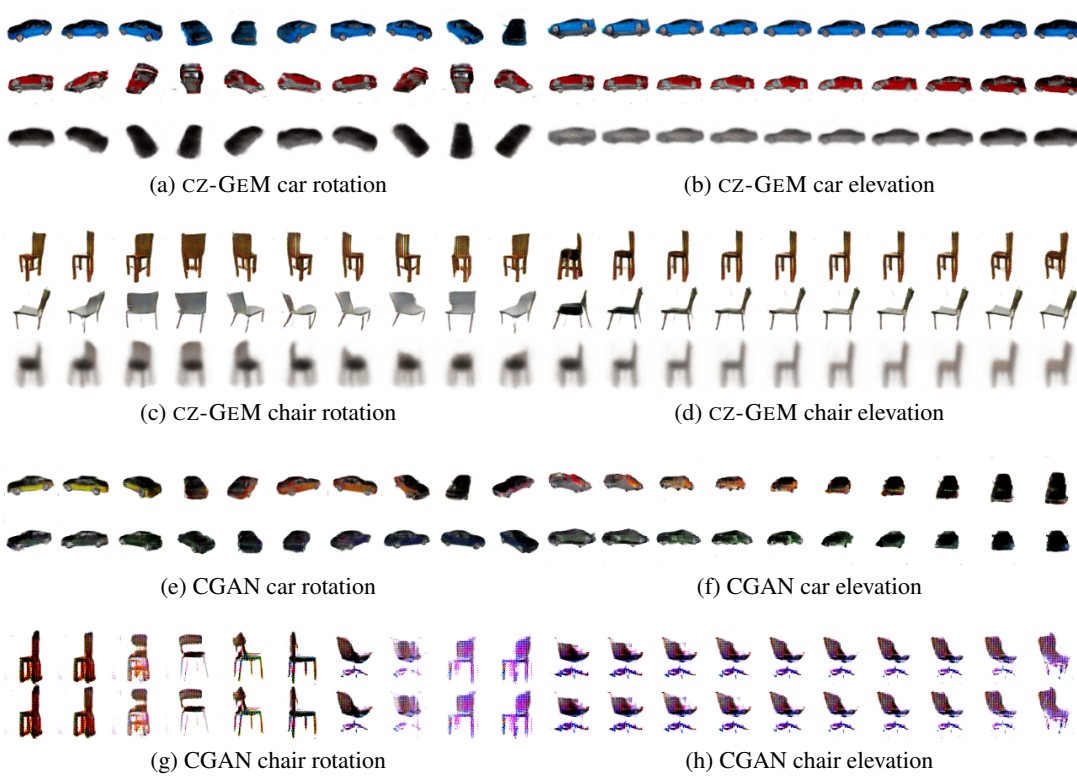

(a) CZ-GEM car rotation      (b) CZ-GEM car elevation

(c) CZ-GEM chair rotation      (d) CZ-GEM chair elevation

(e) CGAN car rotation      (f) CGAN car elevation

(g) CGAN chair rotation      (h) CGAN chair elevation

Figure 4: Latent traversal on cars and chairs. Third rows in CZ-GEM results show $Y$.

We also qualitatively evaluate CZ-GEM on the more challenging faces dataset that includes 10 control variates. As shown in Figure 9 in the appendix, CZ-GEM is not only able to model the common pose factors such as rotation and azimuth but also accurately captures the principal shape component of

Basel face model that approximates the width of the forehead, the width of jaw etc. Compared to CGAN, CZ-GEM does a qualitatively better job at keeping the identity constant.

Finally, in order to ensure that our method does not compromise the generative quality, we evaluate the Inception score (Salimans et al., 2016) on all three datasets. Inception score has been widely used to measure the diversity and the generative quality of GANs. As shown in Table 2, unlike CGAN, CZ-GEM does not degrade the image quality.

### 5.2 UNSUPERVISED SETTING

We now test the performance of CZ-GEM in the unsupervised setting, where disentangled components of $C$ needs to be discovered, using $\beta$-VAE, as part of learning the mapping $C \rightarrow Y$. For our purpose, we use a simple version of the original $\beta$-VAE method with a very narrow bottleneck (6D for faces and 2D for cars and chairs) to extract $C$.

The latent traversals for the faces dataset are presented in Figure 5. Unsupervised discovery is able to recover rotation as well as translation variation present in the dataset. For comparison, we evaluate InfoGAN (Chen et al., 2016) and present the results in Figure 6 where it is evident that CZ-GEM clearly outperforms InfoGAN on both disentanglement and generative quality. More traversal results are provided in the appendix. We further test our method on the CelebA dataset (Liu et al., 2015), where pose information is not available. This traversal plot is shown in Figure 7. Traversal plots for cars and chairs dataset are provided in the Appendix Figure 12 and Figure 13.

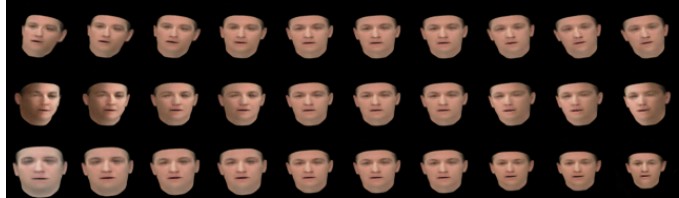

Figure 5: Latent traversal on faces (unsupervised CZ-GEM). The three latent variables capture the rotation, azimuth, and distance respectively.

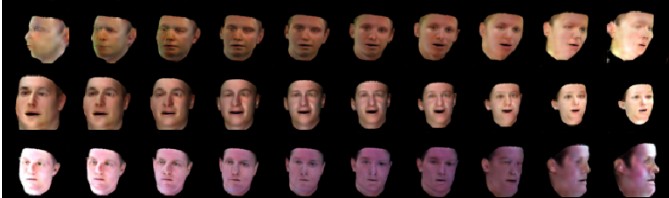

Figure 6: Latent traversal of InfoGAN on faces. The latent variables are able to capture some pose changes but the pose changes are highly entangled with other pose factors as well as the face shape.

## 6 CONCLUSIONS

We present a simple yet effective method of learning representations in deep generative models in the setting where the observation is determined by control variate $C$ and noise variate $Z$. Our method ensures that in the learned representation both $C$ and $Z$ are disentangled as well as the components of $C$ themselves. This is done without compromising the quality of the generated samples. In future work, we would like to explore how this method can be applied to input with multiple objects.

## REFERENCES

Mathieu Aubry, Daniel Maturana, Alexei A Efros, Bryan C Russell, and Josef Sivic. Seeing 3d chairs: exemplar part-based 2d-3d alignment using a large dataset of cad models. In *Proceedings of the IEEE conference on computer vision and pattern recognition*, pp. 3762–3769, 2014.

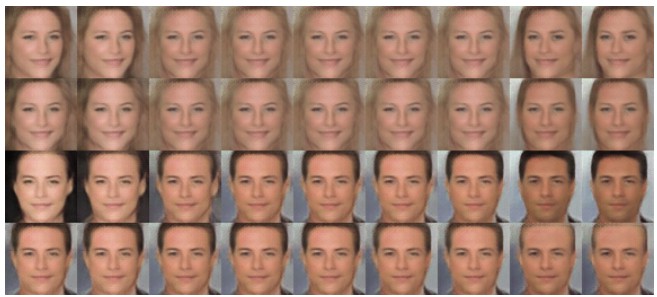

Figure 7: Latent traversal on CelebA (unsupervised CZ-GEM). The latent variables consistently capture the azimuth, hair-style, gender and hair color respectively while maintaining good image quality.

Jianmin Bao, Dong Chen, Fang Wen, Houqiang Li, and Gang Hua. Towards open-set identity preserving face synthesis. In *Proceedings of the IEEE Conference on Computer Vision and Pattern Recognition*, pp. 6713–6722, 2018.

Voler Blanz and Thomas Vetter. A morphable model for the synthesis of 3D faces. In *ACMTOGSIG-GRAPH*, pp. 187–194, 1999.

Andrew Brock, Jeff Donahue, and Karen Simonyan. Large scale gan training for high fidelity natural image synthesis. *arXiv preprint arXiv:1809.11096*, 2018.

Christopher P Burgess, Irina Higgins, Arka Pal, Loic Matthey, Nick Watters, Guillaume Desjardins, and Alexander Lerchner. Understanding disentangling in $\beta$-vae. *arXiv preprint arXiv:1804.03599*, 2018.

Angel X Chang, Thomas Funkhouser, Leonidas Guibas, Pat Hanrahan, Qixing Huang, Zimo Li, Silvio Savarese, Manolis Savva, Shuran Song, Hao Su, et al. Shapenet: An information-rich 3d model repository. *arXiv preprint arXiv:1512.03012*, 2015.

Tian Qi Chen, Xuechen Li, Roger B Grosse, and David K Duvenaud. Isolating sources of disentanglement in variational autoencoders. In *Advances in Neural Information Processing Systems*, pp. 2610–2620, 2018.

Xi Chen, Yan Duan, Rein Houthooft, John Schulman, Ilya Sutskever, and Pieter Abbeel. Infogan: Interpretable representation learning by information maximizing generative adversarial nets. In *Advances in neural information processing systems*, pp. 2172–2180, 2016.

Bernhard Egger, Sandro Schönborn, Andreas Schneider, Adam Kortylewski, Andreas Morel-Forster, Clemens Blumer, and Thomas Vetter. Occlusion-aware 3d morphable models and an illumination prior for face image analysis. *IJCV*, 2018.

Bernhard Egger, William AP Smith, Ayush Tewari, Stefanie Wuhrer, Michael Zollhoefer, Thabo Beeler, Florian Bernard, Timo Bolkart, Adam Kortylewski, Sami Romdhani, et al. 3d morphable face models–past, present and future. *arXiv preprint arXiv:1909.01815*, 2019.

Babak Esmaeili, Hao Wu, Sarthak Jain, Alican Bozkurt, Narayanaswamy Siddharth, Brooks Paige, Dana H Brooks, Jennifer Dy, and Jan-Willem van de Meent. Structured disentangled representations. *arXiv preprint arXiv:1804.02086*, 2018.

Thomas Gerig, Andreas Morel-Forster, Clemens Blumer, Bernhard Egger, Marcel Lüthi, Sandro Schönborn, and Thomas Vetter. Morphable face models - an open framework. In *FG*, pp. 75–82, 2018.

Ian J. Goodfellow, Jean Pouget-Abadie, Mehdi Mirza, Bing Xu, David Warde-Farley, Sherjil Ozair, Aaron C. Courville, and Yoshua Bengio. Generative adversarial nets. In *Neural Information Processing Systems*, 2014.

Michael Gutmann and Aapo Hyvärinen. Noise-contrastive estimation: A new estimation principle for unnormalized statistical models. In *Proceedings of the Thirteenth International Conference on Artificial Intelligence and Statistics*, pp. 297–304, 2010.

Irina Higgins, Loic Matthey, Arka Pal, Christopher Burgess, Xavier Glorot, Matthew Botvinick, Shakir Mohamed, and Alexander Lerchner. beta-vae: Learning basic visual concepts with a constrained variational framework. *ICLR*, 2(5):6, 2017.

Xun Huang and Serge Belongie. Arbitrary style transfer in real-time with adaptive instance normalization. In *Proceedings of the IEEE International Conference on Computer Vision*, pp. 1501–1510, 2017.

Sergey Ioffe and Christian Szegedy. Batch normalization: Accelerating deep network training by reducing internal covariate shift. *arXiv preprint arXiv:1502.03167*, 2015.

Phillip Isola, Jun-Yan Zhu, Tinghui Zhou, and Alexei A Efros. Image-to-image translation with conditional adversarial networks. In *Proceedings of the IEEE conference on computer vision and pattern recognition*, pp. 1125–1134, 2017.

Insu Jeon, Wonkwang Lee, and Gunhee Kim. Ib-gan: Disentangled representation learning with information bottleneck gan. 2018.

Ananya Harsh Jha, Saket Anand, Maneesh Singh, and VSR Veeravasarapu. Disentangling factors of variation with cycle-consistent variational auto-encoders. In *European Conference on Computer Vision*, pp. 829–845. Springer, 2018.

Hyunjik Kim and Andriy Mnih. Disentangling by factorising. *arXiv preprint arXiv:1802.05983*, 2018.

Diederik P. Kingma and Max Welling. Auto-encoding variational bayes. In *2nd International Conference on Learning Representations, ICLR*, 2014.

Durk P Kingma and Prafulla Dhariwal. Glow: Generative flow with invertible 1x1 convolutions. In *Advances in Neural Information Processing Systems*, pp. 10215–10224, 2018.

Adam Kortylewski, Bernhard Egger, Andreas Schneider, Thomas Gerig, Andreas Morel-Forster, and Thomas Vetter. Analyzing and reducing the damage of dataset bias to face recognition with synthetic data. In *CVPRW*, 2019.

Tejas D Kulkarni, William F Whitney, Pushmeet Kohli, and Josh Tenenbaum. Deep convolutional inverse graphics network. In *Advances in neural information processing systems*, pp. 2539–2547, 2015.

Ziwei Liu, Ping Luo, Xiaogang Wang, and Xiaoou Tang. Deep learning face attributes in the wild. In *Proceedings of International Conference on Computer Vision (ICCV)*, December 2015.

Francesco Locatello, Stefan Bauer, Mario Lucic, Sylvain Gelly, Bernhard Schölkopf, and Olivier Bachem. Challenging common assumptions in the unsupervised learning of disentangled representations. *arXiv preprint arXiv:1811.12359*, 2018.

Michael F Mathieu, Junbo Jake Zhao, Junbo Zhao, Aditya Ramesh, Pablo Sprechmann, and Yann LeCun. Disentangling factors of variation in deep representation using adversarial training. In *Advances in Neural Information Processing Systems*, pp. 5040–5048, 2016.

Mehdi Mirza and Simon Osindero. Conditional generative adversarial nets. *arXiv preprint arXiv:1411.1784*, 2014.

Shakir Mohamed and Balaji Lakshminarayanan. Learning in implicit generative models. *arXiv preprint arXiv:1610.03483*, 2016.

Siddharth Narayanaswamy, T Brooks Paige, Jan-Willem Van de Meent, Alban Desmaison, Noah Goodman, Pushmeet Kohli, Frank Wood, and Philip Torr. Learning disentangled representations with semi-supervised deep generative models. In *Advances in Neural Information Processing Systems*, pp. 5925–5935, 2017.

Thu Nguyen-Phuoc, Chuan Li, Lucas Theis, Christian Richardt, and Yong-Liang Yang. Hologan: Unsupervised learning of 3d representations from natural images. *arXiv preprint arXiv:1904.01326*, 2019.

Sebastian Nowozin, Botond Cseke, and Ryota Tomioka. f-gan: Training generative neural samplers using variational divergence minimization. In *Advances in neural information processing systems*, pp. 271–279, 2016.

Alec Radford, Luke Metz, and Soumith Chintala. Unsupervised representation learning with deep convolutional generative adversarial networks. *arXiv preprint arXiv:1511.06434*, 2015.

Scott Reed, Kihyuk Sohn, Yuting Zhang, and Honglak Lee. Learning to disentangle factors of variation with manifold interaction. In *International Conference on Machine Learning*, pp. 1431–1439, 2014.

Tim Salimans, Ian Goodfellow, Wojciech Zaremba, Vicki Cheung, Alec Radford, and Xi Chen. Improved techniques for training gans. In *Advances in neural information processing systems*, pp. 2234–2242, 2016.

Vincent Sitzmann, Michael Zollhöfer, and Gordon Wetzstein. Scene representation networks: Continuous 3d-structure-aware neural scene representations. In *Advances in Neural Information Processing Systems*, 2019.

Akash Srivastava, Lazar Valkov, Chris Russell, Michael U Gutmann, and Charles Sutton. Veegan: Reducing mode collapse in gans using implicit variational learning. In *Advances in Neural Information Processing Systems*, pp. 3308–3318, 2017.

Akash Srivastava, Kai Xu, Michael U Gutmann, and Charles Sutton. Ratio matching mmd nets: Low dimensional projections for effective deep generative models. *arXiv preprint arXiv:1806.00101*, 2018.

Akash Srivastava, Kristjan Greenewald, and Farzaneh Mirzazadeh. Bregmn: scaled-bregman generative modeling networks. *arXiv preprint arXiv:1906.00313*, 2019a.

Akash Srivastava, Jessie Rosenberg, Dan Gutfreund, and David D Cox. Simvae: Simulator-assisted training for interpretable generative models. 2019b.

Masashi Sugiyama, Taiji Suzuki, and Takafumi Kanamori. *Density ratio estimation in machine learning*. Cambridge University Press, 2012.

Attila Szabó, Qiyang Hu, Tiziano Portenier, Matthias Zwicker, and Paolo Favaro. Challenges in disentangling independent factors of variation. *arXiv preprint arXiv:1711.02245*, 2017.

Luan Quoc Tran, Xi Yin, and Xiaoming Liu. Representation learning by rotating your faces. *IEEE transactions on pattern analysis and machine intelligence*, 2018.

Dmitry Ulyanov, Andrea Vedaldi, and Victor Lempitsky. Instance normalization: The missing ingredient for fast stylization. *arXiv preprint arXiv:1607.08022*, 2016.

## A APPENDIX

### A.1 CGAN MUTUAL INFORMATION DERIVATION

Following Sugiyama et al. (2012); Gutmann & Hyvärinen (2010); Mohamed & Lakshminarayanan (2016); Srivastava et al. (2017) we know that at optima the **logits** ($D_\omega$) of a trained discriminator approximate the log-ratio of the true data and generated data densities, i.e. $D_\omega \approx \log \frac{p_x(x|c)p(c)}{p_\theta(x|c)p(c)}$. Following Nowozin et al. (2016); Srivastava et al. (2017; 2018; 2019a) the generator $G_\theta$ can therefore be trained to minimize the following f-divergence between the two sets of densities,

$$\min_\theta \mathbb{E}[-D_\omega] \approx \min_\theta \int p_\theta(x|c)p(c)\left[ -\log \frac{p(x|c)p(c)}{p_\theta(x|c)p(c)}\right] d(x,c). \qquad (2)$$

The RHS of equation 2 can be re-arranged into terms containing the upper and the lower bounds to the generative mutual information between $X$ and $C$, i.e.,

$$
\begin{aligned}
\min_\theta \mathbb{E}\big[-D_\omega\big] &\approx \min_\theta \mathbb{E}\left[\log\frac{p_\theta(x|c)}{p(x)}\right] + \max_\theta \mathbb{E}\left[\log\frac{p(c|x)}{p(c)}\right] \\
&= \min_\theta \mathbb{E}\left[\log\frac{p_\theta(x|c)}{p_\theta(x)} + \log\frac{p_\theta(x)}{p(x)}\right] + \\
&\quad \max_\theta \mathbb{E}\left[\log\frac{q(c|x,\theta)}{p_(c)} + \log\frac{p(c|x)}{q(c|x,\theta)}\right] \\
&= \min_\theta \mathbb{E}\left[\log\frac{p_\theta(x|c)}{p_\theta(x)} + \log\frac{p_\theta(x)}{p(x)}\right] + \\
&\quad \max_\theta \mathbb{E}\left[\log\frac{q(c|x,\theta)}{p(c)} - \log\frac{q(c|x,\theta)}{p(c|x)}\right] \\
&= \min_\theta \mathcal{I}_{g,\theta}(x,c) + \mathbb{E}_{q(c|x,\theta)}\mathrm{KL}(p_\theta(x)\|p(x)) + \\
&\quad \max_\theta \mathcal{I}_{g,\theta}(x,c) - \mathbb{E}_{p_\theta(x)}\mathrm{KL}[q(c|x,\theta)\|p(c|x)] \\
&= \min_\theta \mathcal{I}_{g,\theta}^{UB}(x,c) - \mathcal{I}_{g,\theta}^{LB}(x,c).
\end{aligned}
\tag{3}
$$

## A.2 Quantitative Evaluation

Apart from the MSE-based estimator reported in 1, we report and additional evaluation measure. We use the same regressor $f(x)$ trained for 1, but we report the Pearson correlation co-efficient (r) between the predicted label and the true label $r(c, f(G_\theta(c,z)))$ for each dimension of $C$.

**Chairs**

| CGAN | 0.89 | 0.88 |
|---|---|---|
| CZ-GEM | 0.82 | 0.89 |

**Cars**

| CGAN | 0.88 | 0.97 |
|---|---|---|
| CZ-GEM | 0.86 | 0.96 |

**Faces**

| CGAN | 0.46 | 0.55 | 0.41 | 0.40 | 0.88 | 0.91 | 0.77 | 0.93 | 0.87 | 0.97 |
|---|---|---|---|---|---|---|---|---|---|---|
| CZ-GEM | 0.27 | 0.32 | 0.31 | 0.24 | 0.88 | 0.87 | 0.72 | 0.89 | 0.78 | 0.96 |

## A.3 Additional experiment results

Comparison of CZ-GEM and CGAN on face dataset is shown in Figure 9. CGAN not only produces blurry faces but also shows more undesired identity changes. In order to show the shape variation clearly, we provide a zoomed-in view in Figure 10.

We provide additional results for supervised and unsupervised results on the chair dataset from Aubry et al. (2014) in Figure 11 and Figure 12 respectively. The observation is the same with the previous one. CZ-GEM varies the control variables without changing the shape of chairs. In the first row in Figure 11, the leg of the chairs are visually indistinguishable showing an excellent disentanglement between $C$ and $Z$. For the results in unsupervised setting showing in Figure 12, CZ-GEM is able to disentangle the rotation of chairs without any label.

Additional results of latent traversal of CZ-GEM in the unsupervised setting is provided in Figure 13. The model is able capture the rotation but the translation is not very smooth.

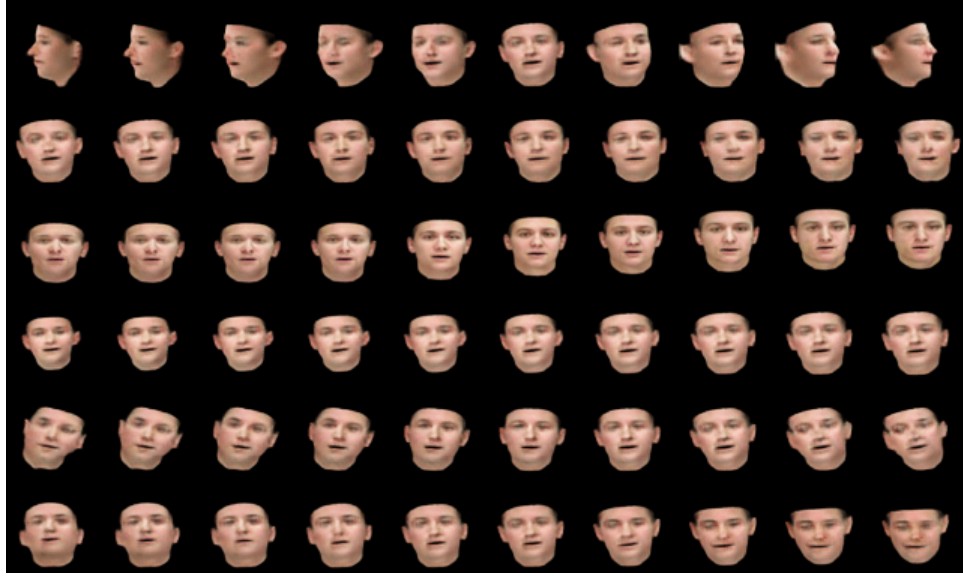

(a) CZ-GEM face pose

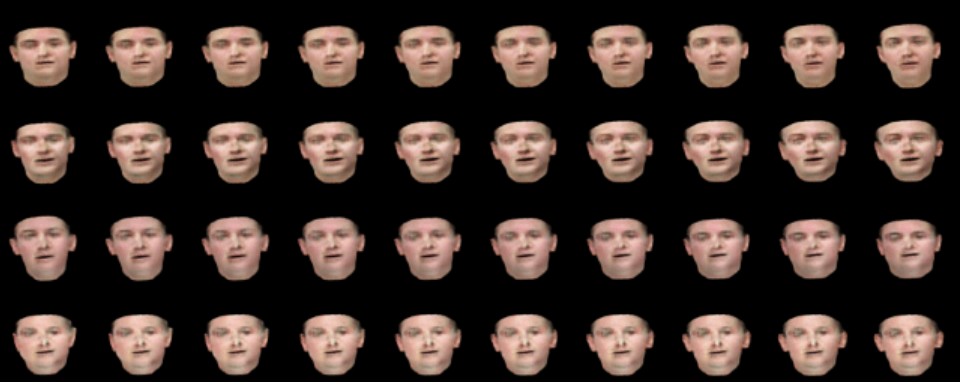

(b) CZ-GEM face shape

Figure 8: Latent traversal of CZ-GEM on faces. The pose variations are azimuth, horizontal translation, vertical translation, distance, rotation, and elevation from top to bottom. The shape variations show the difference in face height, forehead, jaw, and ear from top to bottom.

Figure 14 provides the InfoGAN result on the face dataset. Compared with unsupervised CZ-GEM result in Figure 15, clearly InfoGAN discovers some control variables but the effect is highly entangled.

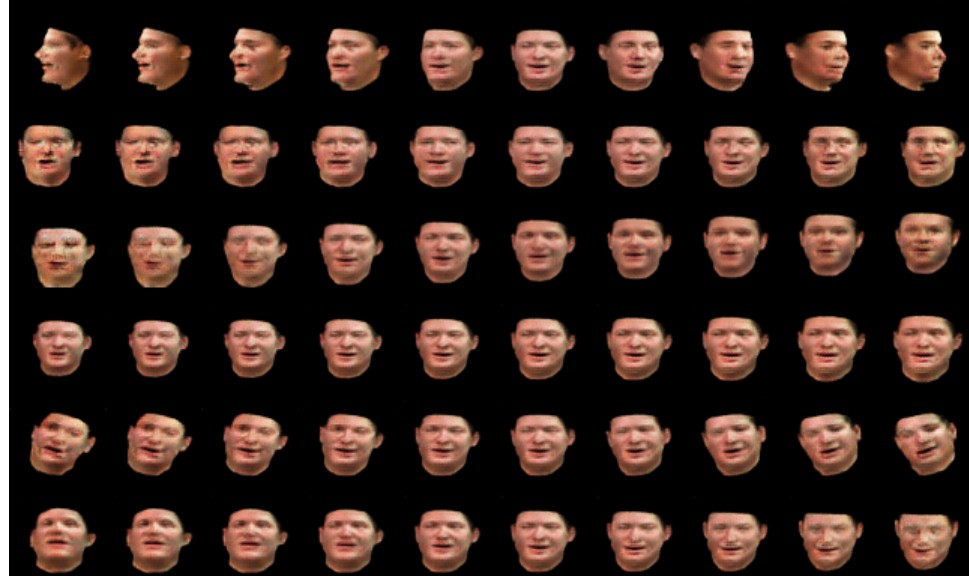

(a) CGAN face pose

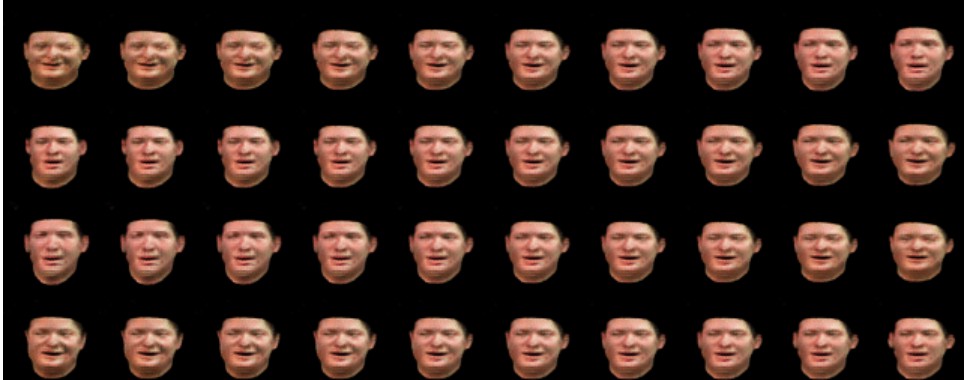

(b) CGAN face shape

Figure 9: Latent traversal of CGAN on faces. The pose variations are azimuth, horizontal translation, vertical translation, distance, rotation, and elevation from top to bottom. The shape variations show the difference in face height, forehead, jaw, and ear from top to bottom.

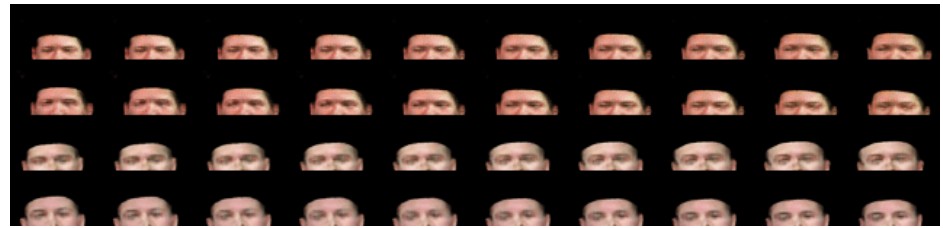

Figure 10: Zoomed-in comparison on face shape. Row 1: CGAN forehead variation; Row 2: CGAN jaw variation; Row 3: CZ-GEM forehead variation; Row 4: CZ-GEM jaw variation. Row 1 and Row 3 should have a bigger forehead from left to right while Row 2 and Row 4 should have a consistent forehead. CGAN and CZ-GEM shows good forehead variation in Row 1 and Row 3 respectively but CZ-GEM (Row 4) does better than CGAN (Row 2) in keeping the forehead the same while another factor is changing.

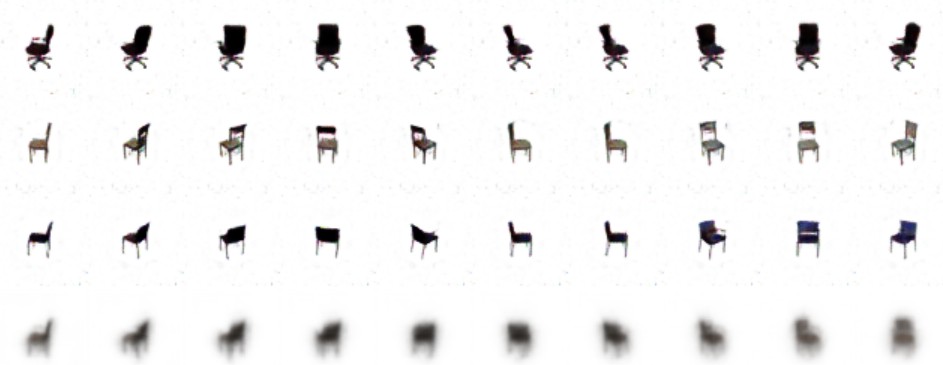

Figure 11: Latent traversal on chairs. The first three rows show the effect of the variable in $C$ that controls the rotation of chairs. The last row visualize the corresponding $Y$.

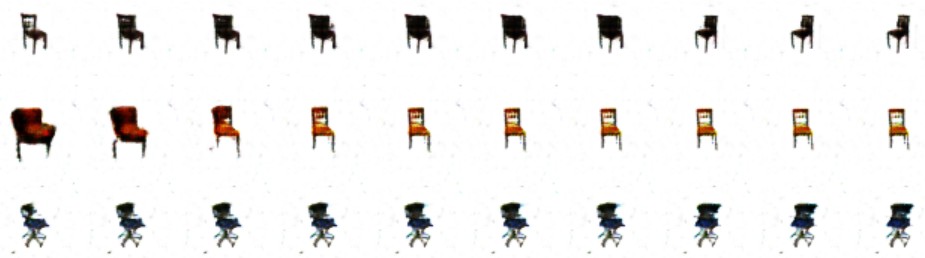

Figure 12: Latent traversal on chairs (unsupervised CZ-GEM).

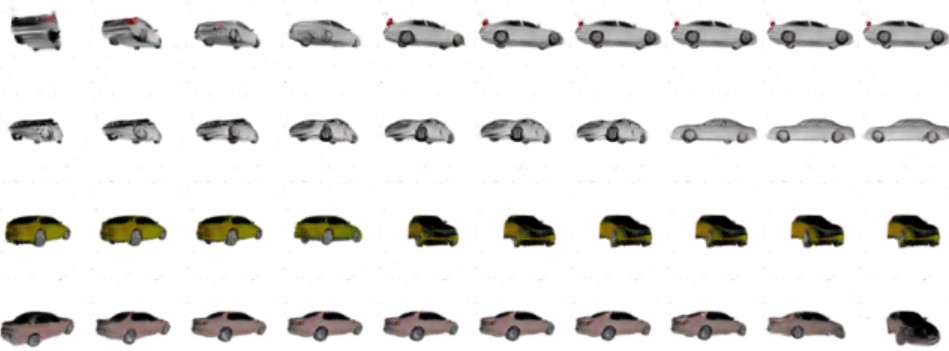

Figure 13: Latent traversal of unsupervised CZ-GEM on cars.

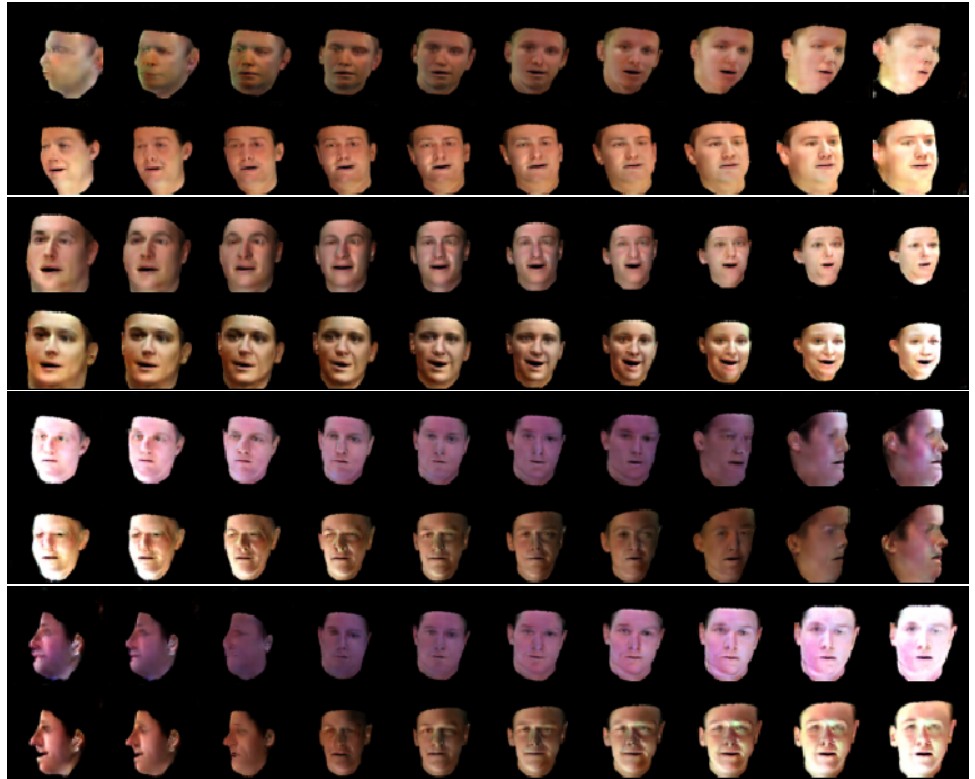

Figure 14: Latent traversal of InfoGAN on faces dataset.

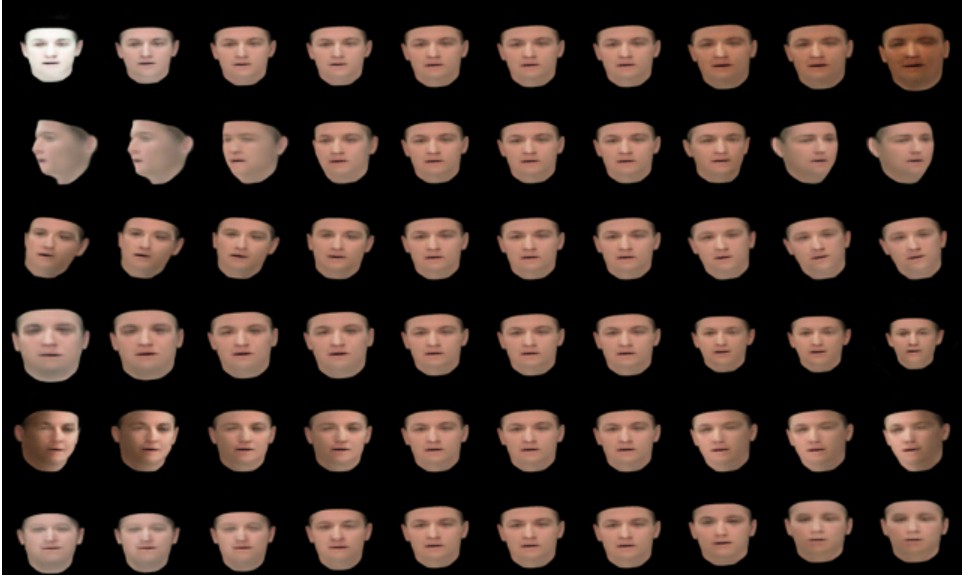

Figure 15: Latent traversal of unsupervised CZ-GEM on face dataset.

