# OpenReview forum: "CZ-GEM:  A  FRAMEWORK  FOR DISENTANGLED REPRESENTATION LEARNING"
_ICLR.cc/2020/Conference — Reject_

### Official Review · AnonReviewer3 · 2019-10-22
**Official Blind Review #3**

**Rating:** 1

**Review:**

This paper proposes a hybrid technique for rendering “control-variate” and class-conditional image in two steps, first by generating an approximate rendering of the image (“Y”) conditional on the control variate and then filling in the details with a conditional GAN dependent on a latent noise variable Z (although I note that the caption of Figure 2 which identifies “Z” as the identity makes this rather confusing).

To ensure that Z is used to explain aspects of the model that are separate from the controlled variation, Z is combined in the refinement model at later steps (since otherwise the posterior over Z and Y conditional on X could induce entanglement between the variables).

In the “supervised” setting where the control variates are observed, Y can be learned as a simple regression problem independent of the other parts of the model, and this two-stage refinement process is demonstrated (using inception scores) to generate convincing samples, including when C consists of up to 10 control variates. In the unsupervised setting, a beta-VAE is used to learn a disentangled representation of X as a proxy for C, but then the data is regenerated using a two step process.

Readability suggestion: the paper starts with a very nice motivating example, but when the setup is provided, i.e., that (x,c) pairs are the input to the learner, the intended content of c is not immediately clear- control variates could assume anything from general context information to privileged information. A similarly informative example would be great!

Clarification regarding lemma 1: it seems that if the true posterior cannot be expressed by q, a gap will necessarily remain, even in the “limit” of perfect learning. Is this correct?

Overall: this paper makes a convincing case that it can be used to generate higher quality images, but not that this improves the quality of the disentangled representations. In fact, the separate training seems to make this unlikely.


**Experience Assessment:**

I have published one or two papers in this area.

**Review Assessment: Checking Correctness Of Derivations And Theory:**

I carefully checked the derivations and theory.

**Review Assessment: Checking Correctness Of Experiments:**

I assessed the sensibility of the experiments.

**Review Assessment: Thoroughness In Paper Reading:**

I read the paper at least twice and used my best judgement in assessing the paper.

---

> ### Author Response · Authors · 2019-11-11
> **We thank reviewer 3 for their feedback.**
>
> -- We will add further clarification regarding what C, Z represent.
>
> -- As rightly mentioned by the reviewer, our method can handle very high dimensional control variates.
>
> -- Lemma 1: Yes, your assumption is correct in general for variational posterior.
>
> -- Improving disentangled representation learning over beta-VAE: Beta-VAE obtains disentangled representations by explicitly posing a trade-off between the ‘quality of disentanglement’ (factorisation of the posterior) vs. the image reconstruction quality. Our method removes this trade-off—-we decouple ‘disentanglement of the latents’ from ‘generation quality’, specifically by having a two-stage training process. This allows us to potentially have much higher disentanglement, while still maintaining image quality, unlike beta-VAE where the quality of generation would necessarily be compromised. We would like to emphasize that this is possible only because of the two-stage training process (please see comments to Reviewer 2 regarding d-separation).

---

### Official Review · AnonReviewer1 · 2019-10-26
**Official Blind Review #1**

**Rating:** 1

**Review:**

Summary:
The paper proposes the use of a hierarchical model for a generative modeling task. They propose a framework of introducing an intermediate latent variable to enforce the independence of the control and noise variable.
The paper report extensive experimental results to validate the proposed hierarchical model.
The authors also provide the anonymized code to observe the exact implementation in TensorFlow to visualize the latent variable traversals.

Comments:
The paper proposes the use of a hierarchical model for a generative modeling task by introducing an intermediate latent variable to enforce the independence of the control and noise variable.
The paper report extensive experimental results to validate the proposed hierarchical model.
This type of framework of crude to fine hierarchical generative model has already been successfully introduced by StackGAN and it's recent variants.
On the unsupervised disentangled feature learning, the framework provides incremental advancement by using beta-VAE in conjunction with GAN to use the best of both the worlds.
Even though the proposed approach is similar to StackGAN, the experiments and the results mentioned in the paper are noteworthy.

Questions to Authors:
There are 2 main claims of novelty made in the paper.
1. Architectural Biases:
How is the approach different in comparison to the StackGAN and it's variable which also use multiple levels of crude to fine image generation?
2. Unsupervised control variable discovery:
This part is just the use of existing disentanglement VAEs to extract the control variables. So how does the paper try to make contributions to improve the disentangled features with the proposed method?
Apart from combining these to existing ideas, what can be considered as an added novelty to improve the quality of the disentangled features?

In summary, I find there is no novelty involved apart from combining the already existing SOTA model in disentangled feature learning (beta-VAE) and image generation (StackGAN).


**Experience Assessment:**

I have published one or two papers in this area.

**Review Assessment: Checking Correctness Of Derivations And Theory:**

N/A

**Review Assessment: Checking Correctness Of Experiments:**

I assessed the sensibility of the experiments.

**Review Assessment: Thoroughness In Paper Reading:**

I read the paper thoroughly.

---

> ### Author Response · Authors · 2019-11-11
> **We thank reviewer 1 for their feedback and suggestions and provide clarification to their questions.**
>
> 1. vs StackGAN: Our method introduces a learning method that allows for training generative models with disentangled latents without compromising on the generative quality (unlike all SOTA mutual information (MI) based disentangled representation learning methods such as beta-VAE or info-GAN). More fundamentally, our method provably avoids issues posed by d-separation that theoretically prohibit disentanglement in the current SOTA methods. This is completely different from the motivation of StackGAN which aims to use iterative refinement (like several other generative models) to learn a generative map from image captions and does not care about disentanglement.
>
> 2. Unsupervised control variable discovery: Beta VAE (or other MI-based methods) disentangle the latents by compromising the generative quality. The more the model forces disentanglement, by giving more weight to a certain information-theoretic regularizer, the worse the generated images become. By decoupling the training into two steps, our method allows for far better disentanglement than beta-VAE like methods without compromising the generative quality.
>
> Novelty: Our method aims to solve the fundamental issue of d-separation in disentangled representation learning. It allows for a theoretically consistent way of obtaining factorisation in the posterior without any information-theoretic penalties. It is true, that one can describe the method as a (non-trivial) combination of beta-vae + GAN. But this description mischaracterizes the fundamental problem that we have identified and proposed a solution for. (Please refer to comments for Reviewer 2 under ‘Scientific Contribution’)

---

### Official Review · AnonReviewer2 · 2019-10-28
**Official Blind Review #2**

**Rating:** 3

**Review:**

This paper proposes a method for learning disentangled representations.  The approach is used on both supervised (where the factors to be disentangled are known) and unsupervised settings. The authors demonstrate the efficacy of their approach in both settings on several datasets with both quantitative and qualitative results.

This task is an important one. However, I found that the contribution of this paper is fairly small. The proposed approach seems reasonable but it is mostly a work of engineering and provides little insights into the problem nor the proposed model.

The setup where labeled data (c) also seems a bit unnatural (this also seems to be confirmed by the fact that the authors had to build datasets for the problem). Perhaps the authors could give examples of situations where this would naturally arise. In practice, it seems difficult to obtain these data for all required variables to be disentangled.

The unsupervised results are more interesting but not very much explored (a single set of sampled faces). I was also curious as to why the learned Y's are blurry. This sort of two-stage generation is also potentially interesting, I was wondering if the authors had ideas to generalize this idea.

I also was not convinced by the experiments which are mostly qualitative. I did not find that this set of experiments provide enough support to the proposed method.


Detailed comments:
- It is a bit unclear to me how the authors propose to obtain independent posteriors over z and c. Is it purely empirical or is there a formal reason that guarantees it?

- Some of the figures your report are compelling but it is a bit unclear to the reader if the results are general (e.g., the examples could have been hand-picked). Are there any quantitative measures you could provide (in addition to Tables 1 and 2 which don't measure the quality of the approach)?

- Comparing to CGAN seems reasonable but given the task at hand, it seems like other methods could have been tried (although I do realize that no one may have done this before for deep generative models).



Other comments:
- In Figure 3, it would be good to label the upper trapezoid.

- Some paragraphs are very long and the manuscript may benefit from segmenting them into multiple paragraphs.

**Experience Assessment:**

I do not know much about this area.

**Review Assessment: Checking Correctness Of Derivations And Theory:**

N/A

**Review Assessment: Checking Correctness Of Experiments:**

I assessed the sensibility of the experiments.

**Review Assessment: Thoroughness In Paper Reading:**

I read the paper at least twice and used my best judgement in assessing the paper.

---

> ### Author Response · Authors · 2019-11-11
> **We thank reviewer 2 for their valuable feedback.**
>
> We would like to start by clarifying the difference between the final implementation (what the reviewer referred to as engineering contribution) of our method with its scientific contribution.
>
> 1. Scientific Contribution: Most recent work on disentangling generative modelling tries to obtain an independent/factorised posterior over the latent generative factors without directly addressing the problem of d-separation, which theoretically prohibits factorisation of the posterior in models such as beta-VAE, conditional GAN or stack GAN.
> To further elaborate, due to d-separation, models from prior work that have the same underlying plate notation either fail to disentangle the representations (since $p(c,z|x) \neq p(c|x)p(z|x)$ ) ) or do so at the cost of lower generative quality—-because their training relies on having an additive information-theoretic penalty term.
> Our method, on the other hand, decouples the problem of learning disentangled latent representations and high fidelity generation into two separate problems by introducing a hierarchical structure (sub-graph c-y) that is trained separate from the rest of the model. This allows obtaining a posterior $p(c|y)p(z|x,y)p(y|x)$, which in fact guarantees the disentanglement of the factors c from z while preserving the generative strength of the model.
>
> 2. Supervised Setting: We would argue that the setting where labelled data (C) is available is more natural than the unsupervised setting as we aim to learn physical simulators (such as graphics engines) that have a well-defined control variate structure. This setup appears in many previous works, e.g. conditional GAN and its derivatives. Testing such models on synthetic datasets (i.e. outputs of graphics engines) where one can control the generative variables is a standard practice in the field and allows for better testing.
>
> 3. Unsupervised results: For the unsupervised setting, in addition to our face dataset and CelebA, we also present the results on the chairs and cars in the Appendix (See Figure 5, Figure 7, Figure 12, Figure 13).
>
> 4. Experiments: We would argue that our qualitative plots and quantitative metrics are in line with the evaluation used in current SOTA work. In fact, we provide a very thorough mix of quantitative and qualitative experiments for both supervised and unsupervised settings. We would like to point out that there are no accepted measures in the field for the quality of learned disentangled representation (see Locatello et. al. [https://arxiv.org/pdf/1811.12359.pdf](https://arxiv.org/pdf/1811.12359.pdf)) and most previous papers in the field include a similar mix of quantitative and qualitative results in their experiments section. Also, we provide all the code so that it can be verified that the reported results are not cherry-picked.

---

> > ### Comment · AnonReviewer2 · 2019-11-15
> > **Thank you**
> >
> > Hi,
> >
> > I just wanted to thank the reviewers for their response. It answers my questions and also I found that it provided me with a bit more context regarding this specific field of research (and corrected some of my erroneous assumptions).

---

### Decision · Program_Chairs · 2019-12-19

**Decision:**

Reject

**Comment:**

The paper addresses the problem of learning disentangled representations in supervised and unsupervised settings.

In general, the problem of representation learning in of course a core problem in ICLR. However, in the set-up described by the authors, R2 commented on the the set-up for supervised being a bit unnatural in as detailed labels need to be given (somewhat confusingly, the labels are called control variates in the paper).

Several reviewers commented on the novelty of the paper being on the low side, with R2 commenting the contribution being fairly small, and R3 noting similarities to stackgan.

There were also some comments on quality, and clarity. On the topic of technical quality, R2 did note that the authors present extensive results, but R3 mentions that the case for the disentanglement improving is not sufficiently supported. In terms of clarity, there was some initial confusing about e.g. the inference procedure, though the authors addressed these issues in the discussion.